# REFLECTION WINDOW: TEXT GENERATION WITH SELECTIVE REFINEMENT

## ABSTRACT

The autoregressive approach to text generation in large language models (LLMs), while widely used, is inherently suboptimal due to the lack of a built-in mechanism to perform refinement and/or correction of the generated content. In this paper, we consider optimality in terms of the joint probability over the generated response, when jointly considering all tokens at the same time. We theoretically characterize the potential deviation of the autoregressively generated response from its globally optimal counterpart that is of the same length. Our analysis suggests that we need to be cautious when noticeable uncertainty arises during text generation, which may signal the sub-optimality of the generation history. To address the pitfall of autoregressive text generation, we propose an approach that incorporates a sliding reflection window and a pausing criterion, such that reflection and generation can be carried out interchangeably as the generation proceeds. Our approach utilizes a selective refinement mechanism to strike the balance between efficiency and optimality, and the experimental results demonstrate the effectiveness of our method.

## 1 INTRODUCTION

Large language models (LLMs) have become increasingly popular in both academia and industry due to their remarkable performance across a wide range of applications (Chang et al., 2023; Qin et al., 2024; Wang et al., 2024). Currently, the most commonly used sampling approach for transformer-based model (Vaswani et al., 2017) is the maximization-based autoregressive decoding (Lin et al., 2020), where the next token is selected according to the highest conditional probability. Leveraging flexible neural network architectures, these models have demonstrated significant empirical success in various tasks (Pan et al., 2024; Wu et al., 2023; Yu et al., 2023).

Despite significant progress in recent language text generation, how to approach the optimal text that one can possibly decode from a language model still remains an open question. Autoregressive sampling is limited by its tendency to converge to local solution, which hinders the ability to handle certain complex tasks (Holtzman et al., 2019). Previous works have demonstrated challenges it faces when generating outputs involve deep logical structures (Welleck et al., 2022) and processing long sequences (Wu et al., 2021). Other than the standard autoregressive methods, Holtzman et al. (2019) proposed Top-$p$ sampling (also known as nucleus sampling), a stochastic method that adjusts the next-token set based on the shape of the probability distribution. Alternatively, different from top-$p$ sampling that selects tokens until their cumulative probability exceeds a certain threshold, top-$k$ sampling limits selection to a fixed number of tokens (Radford et al., 2019; Holtzman et al., 2018; Fan et al., 2018). Another empirical technique involves modulating the probability distribution using the temperature hyperparameter to control randomness in the output (Caccia et al., 2018; Fan et al., 2018; Ficler & Goldberg, 2017; Ackley et al., 1985). Additionally, efficient inference with beam search has also been explored in the recent literature (Xie et al., 2024; Zhu et al., 2024; Wei et al., 2024; Yang et al., 2024b).

Other than sampling approaches, previous literature has also explored improving generated contents through self-correction and see if LLMs can refine responses based on their own outputs, leveraging either external or internal feedback (Pan et al., 2023; Shinn et al., 2023; Yao et al., 2022; Madaan et al., 2023; Bai et al., 2022; Ganguli et al., 2023; Chen et al., 2023c; Kim et al., 2023; Tyen et al., 2023). Numerous studies have demonstrated effective self-correction performance through the integration of external feedback generated by the learned critic models (Paul et al., 2023; Akyürek et al., 2023; Welleck et al., 2022; Peng et al., 2023; Gao et al., 2023) or interacting with human or tools (Gou et al.,

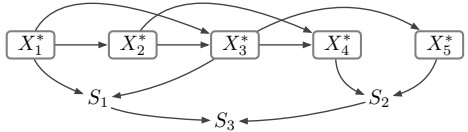
(a) (Sub-)goals jointly satisfied by the optimal sequence

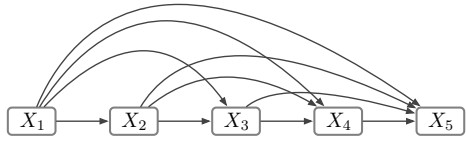
(b) Autogressive way of text generation

Figure 1: Illustrative diagrams of different dependence patterns among variables representing tokens or phrases in text generation. Panel (a): the dependence pattern among variables in the optimal sequence where there are (sub-)goals $S_i$'s to achieve, specifying conditions or constraints that should *jointly* be satisfied by $X_i^*$'s. Panel (b): the autoregressive way of text generation, where $X_i$ is only allowed to depend on $X_j$ if $j < i$.

2023; Chern et al., 2023; Olausson et al., 2023). For better cost efficiency, recent studies also explore approaches to conduct intrinsic self-correction (Li et al., 2024; Shinn et al., 2023; Madaan et al., 2023; Chen et al., 2023b; Manakul et al., 2023; Huang et al., 2023). A related line of research frames text generation as a posterior inference problem, employing sequential Monte Carlo methods for decoding (Lew et al., 2023b;a; Zhi-Xuan et al., 2024). Other works focus on accelerating inference through speculative decoding, where multiple tokens are predicted in parallel (Leviathan et al., 2023; Kim et al., 2024; Chen et al., 2023a; Xia et al., 2024; Sun et al., 2024; Xia et al., 2023).

Previous literature has presented various methods to improve the generation of autoregressive language models. However, the inherent shortcoming of the autoregressive way of generation remains under-explored. In this paper, we theoretically characterize sub-optimality of the autoregressive way of text generation, and propose an empirical method to mitigate this issue. Our contributions can be summarized as follows:

- We theoretically characterize the sub-optimality of autoregressive way of text generation, and demonstrate its inherent shortcoming due to the lack of a built-in mechanism to perform correction/refinement of generated contents.

- We propose a text generation approach involving a sliding reflection window and a corresponding pause criterion, so that reflection and generation can be carried out interchangeably.

- We present empirical evaluations of our proposed approach. We show that the reflection window strategy achieves significant improvement over greedy decoding in inference-intensive settings and maintains performance that is comparable or superior to beam search while being more efficient.

## 2  MOTIVATIONS AND HIGH-LEVEL ILLUSTRATION OF OUR APPROACH

In this section, we first present our motivations behind addressing the inherent shortcoming of autoregressive way of text generation (Section 2.1). Then in Section 2.2, we present a high-level summary of our proposed approach involving interchangeably switching between the reflection (on previously generated content) and generation (of the additional new content).

### 2.1  INHERENT SHORTCOMING OF AUTOREGRESSIVE WAY OF TEXT GENERATION

Recent research advances in cognitive linguistics found that language is primarily a tool for communication (for humans) rather than thought (Fedorenko et al., 2024). Motivated by these findings about human language and thinking, we investigate an inherent shortcoming of purely autoregressive way of sampling approach for text generation, namely, the lack of a built-in mechanism for correcting previous errors and/or refining the generated history, and therefore, suboptimality in the generated response.

Let us consider an example of writing a novel. For a long-format writing like novels, outlining (also referred to as plotting) is essential for structuring ideas, planning narratives, and crafting engaging drafts (King, 2000; Serravallo, 2017). Sub-goals refer to relatively small and achievable tasks that guide the author through each stage of the story, for instance, the setting of the circumstance, the element of tension and emotion, the sensory imagination of the scene.

As we illustrate in Figure 1(a), $X_i^*$'s represent words or phrases in the novel, and $S_i$'s represent sub-goals, which may be related in a hierarchical way, for instance, sub-goals within a single scene

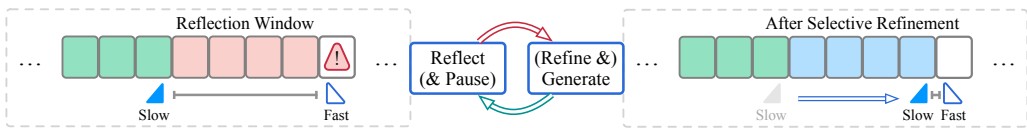

Figure 2: Overview of our approach to address the inherent shortcoming of autoregressive ways of text generation, where the LLM interchangeably switches between reflection and generation. The fast pointer pauses if a certain criterion is triggered, indicating the existence of a potential issue of the generated text. Then, the model refines the generation history between the fast and slow pointers before continuing generation, so that the slow pointer catches up with the fast pointer before the latter can move on. Reflection window refers to the content between the fast and slow pointers.

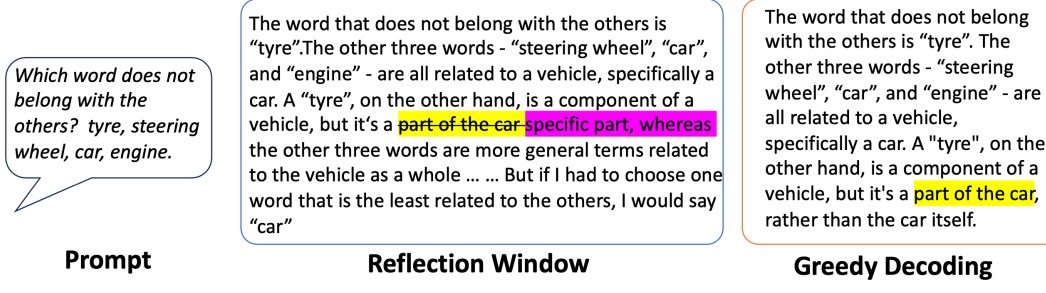

Figure 3: A concrete example demonstrating our propose approach

altogether serve the purpose of furthering the development of the story. We model sub-goals in terms of selection variables $S_i$ since they represent constraints or objectives to achieve, which involve certain criteria to be satisfied over the variables that they operate upon.[1] As we can see from Figure 1(a), the variables in optimal sequence (the novel in this example) $X_i^*$'s *jointly* satisfy criteria, or optimize objectives, specified by sub-goals $S_i$'s. This indicates that the best $X_i^*$ in the optimal sequence depends on best values of all other $X_j^*$'s. However, with an autoregressive way of text generation, as illustrated in Figure 1(b), we only allow $X_i$ to depend on $X_j$'s if $j < i$, which is clearly suboptimal.

## 2.2 Selective Refinement Through Reflection Window

As we have seen in Section 2.1, one inherent shortcoming of the autoregressive way of text generation is the lack of built-in mechanism to correct or refine generated content, so that the overall response satisfies requirements or objectives that operate *jointly* over all involved tokens. In this subsection, we present a high-level summary of our approach as an attempt to address this issue.

Considering the fact that the text typically flows in a single dimension, i.e., from the start to the end, there is a natural linear structure (of words, phrases, and sentences) in text generation. This linear structure differentiates text from other forms of objects that occupy multiple dimensional spaces, e.g., images or videos. Therefore, taking advantage of this linear structure, we propose an approach to perform text generation and reflection at the same time as the output unfolds. There are two different modules: (1) a criterion that specifies whether we should pause the generation since we need to correct/refine generated content, and (2) a mechanism that allows LLMs to always reflect on what was generated and to decide whether a revision is necessary (according to the aforementioned criterion).

As illustrated in Figure 2, given a pausing criterion, we can use fast and slow pointers on the generated content to form segments of a certain length, namely, the sliding reflection window, and perform potential refinements within this sliding window as the text generation proceeds.[2] Notice that the reflection window mechanism does not require the completion of the entire LLM output before the reflection can kick in. This is favorable for several reasons. To begin with, one do not have to wait until the entire response to be finished before starting improving the content. Besides, if there are

---

[1]This modeling choice is consistent with the modeling of causal relations among variables of interest through a directed acyclic graph (DAG) (Spirtes et al., 1993; Pearl, 2009).

[2]The naming of them is motivated by Kahneman (2011).

multiple potential issues in the generation history, the revision can be inefficient if we do not correct them in time, since we need to operate on a more coarse granularity, e.g., by regenerating at the sentence level rather than word or phrase level. Furthermore, because of the linear nature of the text generation, given a pausing criterion, our sliding reflection window mechanism ensures the detection of issues in the generated text, as long as the issue is detectable by the pausing criterion itself.

The empirical pausing criteria we use (Section 5) are guided by our theoretical characterization of the sub-optimality of autoregressive text generation, and to this theoretical analysis we now turn.

## 3 THEORETICAL CHARACTERIZATION OF THE SUB-OPTIMALITY OF AUTOREGRESSIVE TEXT GENERATION

In this section, we theoretically characterize the sub-optimality of autoregressive text generation. We show that even if an LLM is sufficiently trained and can perfectly capture any autoregressive decomposition of the joint density, to maximize the joint probability of the generated text, the autoregressive way of text generation can still deviate from the globally optimal response.

Let us denote a token from the vocabulary $\mathcal{V}$ as $w_v \in \mathcal{V}$, whose index in the vocabulary is $v \in |\mathcal{V}|$. We use "$i : j$" to denote the increasing integer sequence from $i$ to $j$ if $i \leq j$, e.g., $1 : t := 1, 2, \ldots, t$ if $t > 1$, otherwise, $i : j := \varnothing$.

**Definition 3.1** (Stepwise Optimal length-$T$ Response). We say a sequence $w_{\widehat{\mathbf{v}}_T[1]} w_{\widehat{\mathbf{v}}_T[2]} \ldots w_{\widehat{\mathbf{v}}_T[T]}$ is stepwise optimal following the prompt $X_{\leq 0}$, if the sequence consists of tokens that correspond to the highest token-by-token conditional probabilities, denoted by $g(X_t \mid X_{1:t-1}, X_{\leq 0})$ where $t \in [1, T]$:

$$
\begin{aligned}
\widehat{\mathbf{v}}_T[1] &:= \underset{v_1 \in |\mathcal{V}|}{\operatorname{argmax}} \, g(X_1 = w_{v_1} \mid X_{\leq 0}), \\
\widehat{\mathbf{v}}_T[2] &:= \underset{v_2 \in |\mathcal{V}|}{\operatorname{argmax}} \, g(X_2 = w_{v_2} \mid X_1 = w_{\widehat{\mathbf{v}}_T[1]}, X_{\leq 0}), \\
&\ldots \\
\widehat{\mathbf{v}}_T[T] &:= \underset{v_T \in |\mathcal{V}|}{\operatorname{argmax}} \, g(X_T = w_{v_T} \mid X_{1:T-1} = w_{\widehat{\mathbf{v}}_T[1]} w_{\widehat{\mathbf{v}}_T[2]} \ldots w_{\widehat{\mathbf{v}}_T[T-1]}, X_{\leq 0}),
\end{aligned}
\tag{1}
$$

and $\widehat{\mathbf{v}}_T = (\widehat{\mathbf{v}}_T[1], \widehat{\mathbf{v}}_T[2], \ldots, \widehat{\mathbf{v}}_T[T])$.

**Definition 3.2** (Globally Optimal Length-$T$ Response). We say a sequence $w_{\mathbf{v}_T^*[1]} w_{\mathbf{v}_T^*[2]} \ldots w_{\mathbf{v}_T^*[T]}$ is globally optimal among all possible length-$T$ responses following the prompt $X_{\leq 0}$, if it has the highest ground-truth conditional probability, denoted by $f(X_{1:t} \mid X_{\leq 0})$ where $t \in [1, T]$:

$$
\mathbf{v}_T^* = (\mathbf{v}_T^*[1], \mathbf{v}_T^*[2], \ldots, \mathbf{v}_T^*[T]) := \underset{v_i \in |\mathcal{V}|, i=1,2,\ldots,T}{\operatorname{argmax}} f(X_{1:T} = w_{v_1} w_{v_2} \ldots w_{v_T} \mid X_{\leq 0}). \tag{2}
$$

In general, the longer the sequence, the lower the overall probability tends to be. The fair comparison of optimality is length specific, and the optimal response of a shorter length is not necessarily identical to the prefix of the optimal response that is longer in length. For instance, if we were to use 10 words to distinguish between joint and conditional densities, one might say "*joint density combines all variables; conditional adjusts for known outcomes*." However, if we can use 15 words, one might say "*joint density reflects combined probabilities of all variables; conditional density adjusts probabilities given known outcomes*." Therefore, we explicitly keep the length $T$ in the notation of vocabulary indices of tokens that constitute the length-$T$ responses.

**Assumption 3.3** (Oracle LLM). We say an autoregressive LLM is an oracle LLM, if the following relation holds for any response of a length $T \geq 1$:

$$
f(X_{1:T} \mid X_{\leq 0}) = \Pi_{t=1}^{T} g(X_t \mid X_{1:t-1}, X_{\leq 0}). \tag{3}
$$

Assumption 3.3 specifies that after given the prompt or generated text history $X_{\leq 0}$, an oracle (or very well-trained) LLM can recover the ground-truth probability of $X_{1:T}$ as a whole follows $X_{\leq 0}$, by multiplying token-by-token generating probabilities in an autoregressive way.[3] We would like to note that Assumption 3.3 only states that an oracle LLM can perfectly capture the autoregressive way

---

[3]Here, we implicitly assume that the context length of the LLM is sufficiently large.

of probability partitioning of text sequences, and this itself does *not* guarantee the equivalence of stepwise optimal response and the same-length globally optimal response for $T > 1$.[4]

**Assumption 3.4** (Strict Preference Among Same-Length Sequences). For any two length-$T$ different sequences following the prompt $X_{\leq 0}$, there is a strict preference between them in terms of the ground-truth conditional probability $f(X_{1:T} \mid X_{\leq 0})$. The ground-truth conditional probabilities of two length-$T$ sequences equal to each other if and only if the sequences are identical.

Assumption 3.4 specifies that from the ground-truth conditional probability perspective, there is a strict preference between how well two different same-length responses follow the prompt $X_{\leq 0}$, i.e., the ground-truth probability mass function $f(X_{1:T} \mid X_{\leq 0})$ is injective for any given $T > 0$.

**Assumption 3.5** (Irreversible Advantage Once Manifested). When a stepwise optimal length-$T$ response from an oracle (Assumption 3.3) autoregressive LLM $w_{\widehat{\mathbf{v}}_T[1]} w_{\widehat{\mathbf{v}}_T[2]} \ldots w_{\widehat{\mathbf{v}}_T[T]}$ is not the globally optimal length-$T$ response $w_{\mathbf{v}_T^*[1]} w_{\mathbf{v}_T^*[2]} \ldots w_{\mathbf{v}_T^*[T]}$, then if the deviation manifests itself at the length-$L$ ($1 < L \leq T$) prefix-sequences, the advantage of the globally optimal length-$T$ response will not be reversed afterwards:

if $\exists L \in (1, T], f(X_{1:L} = w_{\widehat{\mathbf{v}}_T[1]} w_{\widehat{\mathbf{v}}_T[2]} \ldots w_{\widehat{\mathbf{v}}_T[L]} \mid X_{\leq 0}) < f(X_{1:L} = w_{\mathbf{v}_T^*[1]} w_{\mathbf{v}_T^*[2]} \ldots w_{\mathbf{v}_T^*[L]} \mid X_{\leq 0})$,

then $\forall M \in [L, T], f(X_{1:M} = w_{\widehat{\mathbf{v}}_T[1]} \ldots w_{\widehat{\mathbf{v}}_T[M]} \mid X_{\leq 0}) < f(X_{1:M} = w_{\mathbf{v}_T^*[1]} \ldots w_{\mathbf{v}_T^*[M]} \mid X_{\leq 0})$.

Assumption 3.5 specifies that if the advantage (in terms of a higher ground-truth conditional probability) of the globally optimal length-$T$ sequence can be observed at the length-$L$ prefix-sequence, such advantage will not be reversed when considering longer prefix-sequences.

**Theorem 3.6** (Indication of Deviation from the Globally Optimal Length-$T$ Response). *Given the prompt $X_{\leq 0}$, when an oracle LLM (Assumption 3.3) generates a stepwise optimal length-$T$ response which is not the globally optimal response with the same length, let $L \leq T$ denote the minimum length of prefix-sequence needed in order for such deviation to manifest itself (Assumptions 3.4 and 3.5). Then, the deviation from the globally optimal response happens at some step $K < L$. Furthermore, the conditional probability when generating the token $w_{v_L} \in \mathcal{V}$ is strictly smaller than a positive number, which itself is strictly smaller than 1, i.e.,*

$$1 > \epsilon_L > \max_{w \in \mathcal{V}} g(X_L = w \mid X_{1:L-1} = w_{\widehat{\mathbf{v}}_T[1]} w_{\widehat{\mathbf{v}}_T[2]} \ldots w_{\widehat{\mathbf{v}}_T[L-1]}, X_{\leq 0}),$$

$$where \; \epsilon_L = \frac{f(X_{1:L} = w_{\mathbf{v}_T^*[1]} w_{\mathbf{v}_T^*[2]} \ldots w_{\mathbf{v}_T^*[L-1]} w_{\mathbf{v}_T^*[L]} \mid X_{\leq 0})}{f(X_{1:L-1} = w_{\widehat{\mathbf{v}}_T[1]} w_{\widehat{\mathbf{v}}_T[2]} \ldots w_{\widehat{\mathbf{v}}_T[L-1]} \mid X_{\leq 0})}. \tag{4}$$

Theorem 3.6 provides a necessary (but not sufficient) condition for the deviation of the stepwise optimal length-$T$ response from the same-length globally optimal response. The uncertainty (i.e., low conditional probabilities) in generating the next token can result from different factors. For instance, a previous mistake or detour makes it challenging to continue in any way that could possibly satisfy the goal specified by the prompt. Such uncertainty can also result from multiple valid ways to proceed in order to achieve the goal. Although we do not have access to the ground-truth conditional probability $f(X_{1:T} \mid X_{\leq 0})$, Theorem 3.6 states that when noticeable uncertainty arises, one should to be cautious of a potential deviation from the globally optimal response in the generated text.

# 4 (SEMI)-SYNTHETIC EXPERIMENTS

The principles behind our approach are straightforward; however, one might question whether the observed phenomenon persists in real-world LLM decoding scenarios. To provide clear evidence of the previous theoretical analysis and also address the aforementioned concern, we introduce a (semi)-synthetic experiment designed to demonstrate that, in moderately realistic settings, greedy decoding during text generation with step-wise optimization results in suboptimal responses. Before presenting the experimental results, we first outline the semi-synthetic setup.

**Illustrative Approximation** Although there is no doubt about the existence of an optimal response, for any modern large language model with a vocabulary size $|\mathcal{V}|$ (typically on the order of $10^4$

---

[4]When $T = 1$, i.e., if the response is of a length 1, the stepwise optimal is just the globally optimal for an oracle LLM, since there is only one step in total, and $f(X_1 \mid X_{\leq 0}) = g(X_1 \mid X_{\leq 0})$.

to $10^5$), identifying the joint optimal sequence across multiple steps becomes computationally intractable, even for relatively short sequence lengths ($< 100$). To uphold the validity of our claim while offering an accessible illustration, we adopt beam search as an approximation of the globally optimal sequence. Since we measure the probability that greedy decoding achieves the optimal response, this approximation serves as an upper bound on achievable performance, highlighting the discrepancy between greedy decoding and the true globally optimal response.

**Approximating Natural Language Scenarios**   One may also be curious about the prompt or context in the generation since those factors will influence the generation behavior as well. It is crucial to demonstrate this in realistic scenarios. Specifically, we align our experimental setting with common human interactions with large language models. To achieve this, we utilize MT-Bench questions as curated prompts and contexts, which are designed for evaluating conversational, open-ended chat models. These samples serve as an approximation of real-world natural language context distributions, ensuring that our findings are grounded in practical, human-like use cases.

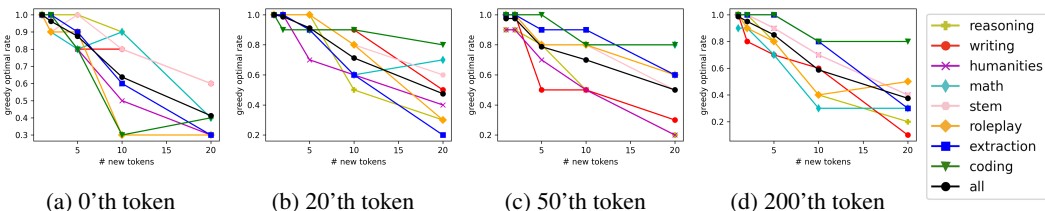

(a) 0'th token        (b) 20'th token        (c) 50'th token        (d) 200'th token

Figure 4: Probability of greedy decoding results in globally-optimal response with respect to the number of newly generated tokens (the legend is shared across figures)

**Results**   For each prompt or context, we evaluate whether the joint probability of the sequence generated through greedy decoding is greater than or equal to that of the sequence produced by the approximate global optimum method, i.e., beam search. This comparison highlights the extent to which greedy decoding diverges from the optimal sequence. As demonstrated in Figure 4, greedy decoding consistently results in suboptimal sequences, even when generating a small number ($< 10$) of new tokens. Additionally, the behavior varies across positions in the generated text. For example, the beginnings of the answers usually diverge, and it can be harder for greedy decoding to achieve optimal results. To reduce potential inductive biases arising from greater diversity at the initial stages of generation, we evaluate sequences at various positions throughout the generated text (Figures 4(a)-4(d) capture different starting positions for the evaluation). Our findings indicate that this behavior persists across all evaluated positions, underscoring the importance of the issue. This observation naturally suggests a solution: a reflective decoding strategy using a sliding window mechanism.

## 5  EXPERIMENTS ON REAL-WORLD TASKS

In this section, we present experimental results of reflection window on two different real-world tasks: MMLU and MT-Bench. The experimental results demonstrate both the effectiveness and efficiency of our proposed method.

### 5.1  EXPERIMENTAL SETUP

**Pausing Criterion**   Guided by our theoretical charactization of the sub-optimality of the autoregressive way of text generation (Theorem 3.6), the pausing criterion in the empirical approach needs to capture the (trend of) increase in uncertainty at the fast pointer (Section 2.2). For an empirical pausing criterion, we use the conditional entropy $H(\cdot)$ based on the next-token logits across the vocabulary. Specifically, the pausing criterion is formulated as follows:

Given an LLM which models the conditional distribution $g(X_t \mid X_{1:t-1})$ of the token in $t$-th step given all the observed history context $X_{1:t-1} = \mathbf{x}_{1:t-i}$, the pausing criterion $h(t; \sigma, d)$ is defined as follows

$$h(t; \sigma, d) = \begin{cases} \text{True} & \text{if } H\big(X_t \mid X_{1:t-i} = \mathbf{x}_{1:t-i}\big) > \sigma \,, \forall i \in [0, d-1], \\ \text{False} & \text{Else.} \end{cases} \tag{5}$$

The entropy threshold $\sigma$ and window size $d$ are hyperparameters of the pausing criterion. When $h(t; \sigma, d)$ is True, it means the current autoregressive generation should pause at $t$-th step and look back into the reflection window for refinement. Those two parameters jointly decide the sensitivity and effective region of the pausing creterion.

**Refinement Process**   When the pausing criterion is met in $t$-th step, the model will be looped back by $d-1$ steps. The model have to regenerate previous $d$ tokens in the reflection window. Since beam search can approximate the global optimum well, especially when the sequence length is fixed, we introduce beam search on the $t - d + 1$-th step to generate a new subsequence with length $d$. After the refinement, the model will shift back to autoregressive generation until another pausing criterion is met.

**Model**   We choose Llama-3.1-8B-Instruct for most parts of the experiments, including quantitative results and part of the analysis. Llama-3.1-8B-Instruct (denoted as Llama3.1-8B) belongs to Llama 3.1 series, which is the updated version of Llama3 family (Dubey et al., 2024).

To further analyse the general applicability of reflection window mechanism, we leverage several popular models across several LLM families: Phi-3-Medium-128K-Instruct (Abdin et al., 2024) (denoted as Phi-3-medium) with 14 billion parameters, Qwen2.5-14B-Instruct (Yang et al., 2024a) (denoted as Qwen2.5-14B) with 14 billion parameters and Mistral-Nemo-Instruct-2407 (MistralAI, 2024) (denoted as Mistral-Nemo) with 12 billion paramters.

**Benchmarks and Evaluation Metrics**   We use MMLU (Hendrycks et al., 2020) and MT-Bench (Zheng et al., 2023) as benchmarks for real-world scenarios. MMLU tests the model's general knowledge across 57 diverse subjects, including humanities, STEM(Science, Technology, Engineering, and Mathematics), and social sciences, at varying difficulty levels, making it a comprehensive measure of the model's reasoning and factual knowledge. MT-Bench, on the other hand, provides a fine-grained evaluation of LLMs through multi-turn dialogue tasks, assessing not just correctness, but also coherence, reasoning ability, and language fluency.

For MMLU, we adopt macro averaged accuracy as performance measurement because the number of problems for each subjects are different. For MT-Bench, we use win rate as the performance metric. For each pair of response for comparison, we prompt the LLM judge with two responses, one in the front and another one later, follow the same prompting method proposed in MT-Bench paper Zheng et al. (2023). The LLM judge should return a decision from three options: win, lose or tie.

However, due to the preference bias to the first candidate, LLM judge tend to select the first response as the better one. For each pair, we prompt GPT-4o twice, so that both responses will be treated equally. If one response get two win judgement or one win and one tie judgement from GPT-4o, we will note this response prevails the other. Similarly, if one response get two lose judgements or one lose and one tie judgement, this response will be marked as lose. The rest situations will all be treated as tie.

We use win rate as the measurement for the performance comparison on MT-Bench as follows:

$$\text{win rate} = \frac{\#\text{Num Wins}}{\#\text{Num Wins} + \#\text{Num Loses}} \tag{6}$$

#Num Wins stands for the number of win response and #Num Loses for the number of lose.

**Baseline Methods**   We compare reflection window with two baseline methods: greedy decoding and full beam search. Greedy decoding is to select the token which maximizes the conditional probability. While the full beam search (denoted as beam search for short) maintains a set of candidate sequences during the whole generation process. We use a beam size of 4 for all the experiments presented within this section. Reflection window only leverages beam search when the pausing criterion $h(t; \sigma, d)$ is true. That is to say, the subsequence under the reflection window requires to be regenerated. To make fair comparison with full beam search, we also set the beam size to 4 for all reflection window implementation in experiments.

Table 1: Accuracy on MMLU by greedy decoding, beam search and reflection window

| Method | STEM | Humanities | Social Science | Other | Average |
|--------|------|-----------|----------------|-------|---------|
| Greedy Decoding | 68.31% | 64.38% | 73.29% | 73.37% | 69.60% |
| Beam Search | **70.22%** | **64.87%** | **75.30%** | **73.97%** | **70.83%** |
| Reflection Window | 69.39% | 64.40% | 73.45% | **73.97%** | 70.05% |

Table 2: Win Rate Against Greedy Decoding on MT-Bench

| Method | Win | Lose | Tie | Win rate |
|--------|-----|------|-----|----------|
| Beam Search | 25 | 29 | 26 | 46.30% |
| Reflection Window | 26 | 13 | 41 | **66.67%** |

## 5.2 QUANTITATIVE RESULTS

In this subsection, we quantitatively show the performance comparison between greedy decoding, beam search and reflection window. The reflection window in this subsection is set to have entropy threshold $\sigma = 0.5$ and window size $d = 4$. To provide a broader scope of evaluation, we choose a close-ended, multiple-choice problem based benchmark: MMLU and an open-ended text generation benchmark: MT-Bench. MMLU focuses on the correctness of reasoning while MT-Bench incorporates text generation quality into evaluation.

**MMLU** We present a quantitative comparison between our proposed Reflection Window method and the greedy decoding algorithm. As shown in Table 1, both methods are based on the Llama3.1-8B-Instruct model and were evaluated across all four category subsets of the MMLU: humanities, STEM, social sciences, and others. It is noteworthy that our method outperformed the greedy decoding in all four category subsets. Specifically, our Reflection Window method achieved a macro-accuracy of 69.39% on the STEM category subset, which is an improvement of 1.06% over greedy decoding. Our method also showed improvements in other category subsets, though the margins were relatively smaller compared to the STEM subset. We attribute this to the nature of STEM category questions, which more rigorously test the model's reasoning abilities, thereby enabling the Reflection Window method to more effectively prevent the model from deviating from its ultimate objective. The full result of all subjects of MMLU can be seen in Table A6 in Appendix.

**MT-Bench** On the MT-Bench dataset, the Reflection Window method clearly outperformed both greedy decoding and beam search. Table 2 gives the overall win rate of beam search and reflection window when compared with greedy decoding. In comparisons with greedy decoding, according to assessments by GPT-4o, reflection window prevails in 66.67% of cases, a win rate significantly higher than that of beam search, which only outperformed greedy decoding in 46.3% of cases. This demonstrates that the Reflection Window method can significantly enhance the reasoning performance of large language models in multi-turn dialogues, and ensure their adherence to instructions within conversations.

As illustrated in Figure 5, reflection window demonstrates superior performance over greedy decoding and beam search in STEM and Math categories. We attribute this to the greater need for self-correction by large language models (LLMs) when handling reasoning-intensive tasks such as STEM and Math, to ensure they maintain a clear logical chain and have a definitive goal. This makes reflection window method more beneficial for such tasks. Furthermore, the enhanced performance of reflection window in the Reasoning category substantiates this argument. Consequently, in tasks like those in the humanities, which demand less reasoning ability and self-correction, the performance of reflection window appears comparable to that of greedy decoding.

## 5.3 ANALYSIS

**Experimental Results on More LLMs** To verify the generalizability of the Reflection Window method and its independence from any specific model architecture, we evaluated our method using a

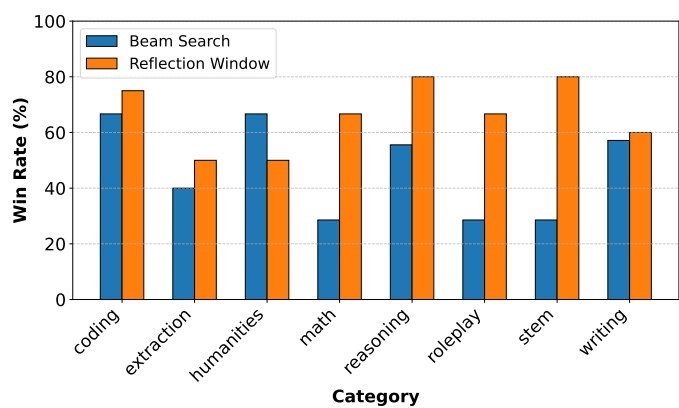

Figure 5: Win Rate Against Greedy Decoding Comparison on MT-Bench by Category

Table 3: Accuracy of Reflection Window on Selected Subsets by Several LLMs

| Model | Method | anatomy | computer_security | global_facts | Average |
|---|---|---|---|---|---|
| Llama3.1-8B | Greedy Decoding | 67.40% | 73.00% | 43.00% | 61.79% |
| | Beam Search | **74.07%** | **78.00%** | 45.00% | 66.57% |
| | Reflection Window | 72.59% | 77.00% | **50.00%** | **67.16%** |
| Phi-3-medium | Greedy Decoding | 73.33% | 70.00% | 56.00% | 66.44% |
| | Beam Search | 70.37% | **78.00%** | 58.00% | 68.79% |
| | Reflection Window | **74.07%** | 76.00% | **62.00%** | **70.69%** |
| Qwen2.5-14B | Greedy Decoding | **74.81%** | **88.00%** | 57.00% | 73.27% |
| | Beam Search | 72.59% | 85.00% | 57.00% | 71.53% |
| | Reflection Window | 74.07% | 85.00% | **63.00%** | **74.02%** |
| Mistral-Nemo | Greedy Decoding | 62.96% | **74.00%** | 45.00% | 60.90% |
| | Beam Search | 60.74% | 65.00% | **53.00%** | 59.70% |
| | Reflection Window | **64.44%** | **74.00%** | 48.00% | **62.39%** |

series of different large language models across three subsets of the MMLU STEM category. Specifically, we additionally selected Phi-3-medium, Qwen2.5-14B, and Mistral-Nemo for experimentation. The experiments are conducted on three selected subsets: "anatomy", "computer_security" from STEM and "global_facts" from Others section of MMLU. The full results are presented in Table 3.

On Phi-3-medium and Mistral-Nemo, our method achieves significant improvements over greedy decoding and beam search, with respective increases of 4.25% and 1.29% compared to greedy decoding. On Qwen2.5-14B, there is also an improvement of 0.75% compared to greedy decoding. These results demonstrate the robustness of the Reflection Window method to changes in model architecture, confirming it as a reliable self-correction mechanism for large language models that does not falter with model alterations. Additionally, we observe that the effectiveness of beam search is not consistently superior to greedy decoding across different large language models; for instance, in Qwen2.5-14B and Mistral-Nemo, beam search performed 1.74% and 1.2% lower than greedy decoding, respectively.

Table 4: Regeneration Measurement by Category on MMLU.

| Category | Regeneration Rate (%) | Num of Regeneration | Num of All Tokens |
|---|---|---|---|
| Humanities | 5.04 | 4.27 | 336.69 |
| Other | 5.54 | 4.31 | 305.93 |
| STEM | 3.50 | 3.15 | 398.55 |
| Social Sciences | 4.82 | 3.84 | 316.29 |

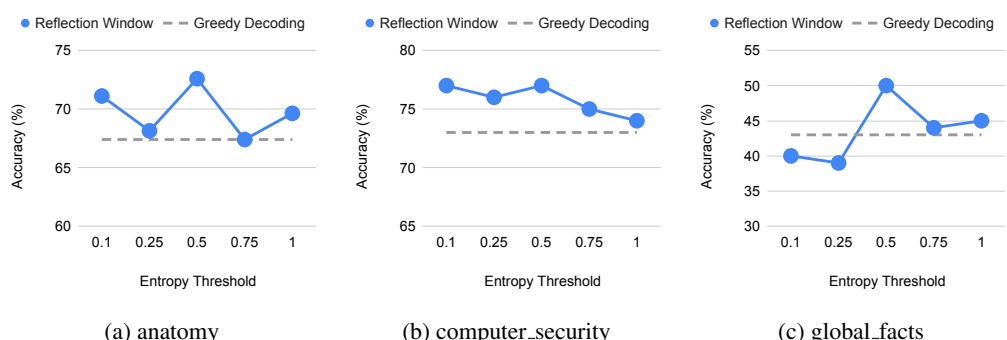

(a) anatomy       (b) computer_security       (c) global_facts

Figure 6: Accuracy on the Selected Subsets with Entropy Threshold $\sigma$

Table 5: Averaged Accuracy on Selected Subsets by Different Entropy Thresholds

| Method | $\sigma = 0.1$ | $\sigma = 0.25$ | $\sigma = 0.5$ | $\sigma = 0.75$ | $\sigma = 1.0$ | *Greedy* |
|---|---|---|---|---|---|---|
| Acc | 63.58% | 61.79% | **67.16%** | 62.69% | 63.58% | 61.79% |

**Parameter Analysis on Entropy Threshold** $\sigma$    In Figure 6, we examined the impact of the hyperparameter: entropy threshold $\sigma$. We choose five distinctive of values between 0 and 1: 0.1, 0.25, 0.5, 0.75, and 1. We choose the same subset from MMLU as Section 5.3.When the threshold is set to 0.5, the Reflection Window method achieves its best performance. Conversely, when the threshold is set to extreme values such as 0.1 or 1.0, the Reflection Window method may degrade to perform similarly to Greedy decoding. It's noteworthy that, the performance trend on these three subjects are different. The reason could probably be the domain shift between different tasks, subjects, etc. It indicates that, although $\sigma = 0.5$ is a relatively safe and stable option for pausing criterion, a more dynamic and adaptive strategy could help to alleviate the effect caused by text domain gap.

**Efficiency of Reflection Window**    We aggregate the regeneration statistics on full MMLU dataset with Llama3.1-8B. We record two key components: number of regeneration - how many times the pausing criterion will be met and start to regenerate within one response; and the second one is the regeneration rate, which characteristics the probability for a token which needs to be modified. As shown in Table 4, the regeneration rate and the number of generation in STEM subsection is much lower than other categories. Since the regeneation rate is between 3.5% and 5.5% across all categories, it means only such small portion of tokens are selected to perform beam search in order to modify. Comparing to beam search search which always keeps a full candidate sequence frontier, reflection window only activate sub-sequential level of beam search when necessary.

## 6   CONCLUSION

In this paper, we theoretically characterize one inherent shortcoming, among others, of the autoregressive way of text generation in LLMs. In particular, we show that when the optimality is defined in terms of the joint probability over all generated tokens, an oracle LLM can still potentially deviate from the globally optimal response of the same length. To mitigate the sub-optimality of the autoregressive way of text generation, we propose an empirical approach guided by our theoretical characterization. We incorporate a sliding reflection window and a pausing criterion so that reflection and generation can be performed interchangeably. Our experimental results demonstrate that the reflection window strategy achieves significant improvement over greedy decoding in inference-intensive settings and maintains performance that is comparable, or even superior to, beam search while being more efficient.

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

# Supplement to
# "Reflection Window: Text Generation with Selective Refinement"

**Anonymous authors**

Table of Contents: Appendix

## A   The Proof of Theoretical Result

**Theorem A.1** (Indication of Deviation from the Globally Optimal Length-$T$ Response)**.** *Given the prompt $X_{\leq 0}$, when an oracle LLM (Assumption 3.3) generates a stepwise optimal length-$T$ response which is not the globally optimal response with the same length, let $L \leq T$ denote the minimum length of prefix-sequence needed in order for such deviation to manifest itself (Assumptions 3.4 and 3.5). Then, the deviation from the globally optimal response happens at some step $K < L$. Furthermore, the conditional probability when generating the token $w_{v_L} \in \mathcal{V}$ is strictly smaller than a positive number, which itself is strictly smaller than $1$, i.e.,*

$$1 > \epsilon_L > \max_{w \in \mathcal{V}} g(X_L = w \mid X_{1:L-1} = w_{\widehat{\mathbf{v}}_T[1]} w_{\widehat{\mathbf{v}}_T[2]} \ldots w_{\widehat{\mathbf{v}}_T[L-1]}, X_{\leq 0}),$$

$$\textit{where } \epsilon_L = \frac{f(X_{1:L} = w_{\mathbf{v}_T^*[1]} w_{\mathbf{v}_T^*[2]} \ldots w_{\mathbf{v}_T^*[L-1]} w_{\mathbf{v}_T^*[L]} \mid X_{\leq 0})}{f(X_{1:L-1} = w_{\widehat{\mathbf{v}}_T[1]} w_{\widehat{\mathbf{v}}_T[2]} \ldots w_{\widehat{\mathbf{v}}_T[L-1]} \mid X_{\leq 0})}. \tag{7}$$

*Proof.* We first show that the deviation from the globally optimal response happens before step $L$. Then, we show that the conditional probability when generating the token $w_{v_L}$ is bounded away from 1.

By definition of oracle LLM (Assumption 3.3), the advantage of the globally optimal response cannot manifest itself at $L = 1$ (even if the deviation happens at step 1), i.e., $L > 1$. Since the minimum length of prefix-sequence needed in order for the deviation of stepwise optimal response from the same-length globally optimal response to manifest is $L$, then the advantage of the globally optimal response is not manifested until step $L$. Until step $L - 1$, in terms of the ground-truth conditional probability following the prompt $X_{\leq 0}$, prefix-sequences of the globally optimal response is *not*

strictly preferred compared to their same-length counterparts of the stepwise optimal response:

$$f(X_1 = w_{\widehat{\mathbf{v}}_T[1]} \mid X_{\leq 0}) \geq f(X_1 = w_{\mathbf{v}_T^*[1]} \mid X_{\leq 0}),$$
$$f(X_{1:2} = w_{\widehat{\mathbf{v}}_T[1]} w_{\widehat{\mathbf{v}}_T[2]} \mid X_{\leq 0}) \geq f(X_{1:2} = w_{\mathbf{v}_T^*[1]} w_{\mathbf{v}_T^*[2]} \mid X_{\leq 0}),$$
$$\dots \tag{8}$$
$$f(X_{1:L-1} = w_{\widehat{\mathbf{v}}_T[1]} w_{\widehat{\mathbf{v}}_T[2]} \dots w_{\widehat{\mathbf{v}}_T[L-1]} \mid X_{\leq 0}) \geq f(X_{1:L-1} = w_{\mathbf{v}_T^*[1]} w_{\mathbf{v}_T^*[2]} \dots w_{\mathbf{v}_T^*[L-1]} \mid X_{\leq 0}).$$

Starting from step $L$ and onwards (Assumption 3.5), prefix-sequences of the globally optimal response are strictly preferred compared to their counterparts of the stepwise optimal response:

$$f(X_{1:L} = w_{\widehat{\mathbf{v}}_T[1]} \dots w_{\widehat{\mathbf{v}}_T[L-1]} w_{\widehat{\mathbf{v}}_T[L]} \mid X_{\leq 0}) < f(X_{1:L} = w_{\mathbf{v}_T^*[1]} \dots w_{\mathbf{v}_T^*[L-1]} w_{\mathbf{v}_T^*[L]} \mid X_{\leq 0}),$$
$$\dots \tag{9}$$
$$f(X_{1:T} = w_{\widehat{\mathbf{v}}_T[1]} w_{\widehat{\mathbf{v}}_T[2]} \dots w_{\widehat{\mathbf{v}}_T[T]} \mid X_{\leq 0}) < f(X_{1:T} = w_{\mathbf{v}_T^*[1]} w_{\mathbf{v}_T^*[2]} \dots w_{\mathbf{v}_T^*[T]} \mid X_{\leq 0}).$$

Assumption 3.4 specifies that for any two same-length but different sequences following the prompt $X_{\leq 0}$, there is a strict ordering between them. Then, in order for the advantage of the globally optimal length-$T$ response to manifest, in terms of strict preferences staring from the length-$L$ prefix-sequence (Equation (9)), there is *at least one* strict preference of the prefix-sequence of stepwise optimal response over its globally optimal counterpart before step $L$. In other words, there is at least one step $K \in [1, L-1]$ such that a strict preference (">" instead of "$\geq$") is present in Equation (8):

$$f(X_{1:K} = w_{\widehat{\mathbf{v}}_T[1]} w_{\widehat{\mathbf{v}}_T[2]} \dots w_{\widehat{\mathbf{v}}_T[K]} \mid X_{\leq 0}) > f(X_{1:K} = w_{\mathbf{v}_T^*[1]} w_{\mathbf{v}_T^*[2]} \dots w_{\mathbf{v}_T^*[K]} \mid X_{\leq 0}). \tag{10}$$

In order to see why this is the case, consider the opposite scenario where there is *no* strict preference in Equation (8). Under Assumption 3.4, the comparison between prefix-sequences is either strict preference (they are different) or exactly the same (identical sequences). If there is no strict preference in Equation (8), then for all $t \in [1, L-1]$, $w_{\widehat{\mathbf{v}}_T[t]} = w_{\mathbf{v}_T^*[t]}$, i.e., the first $L-1$ tokens in the stepwise optimal response are the length-$(L-1)$ prefix of the globally optimal response. If this is the case, the token generated at step $L$ has to deviate from the globally optimal response (since $L$ is the minimum length for the deviation to manifest) $w_{\widehat{\mathbf{v}}_T[L]} \neq w_{\mathbf{v}_T^*[L]}$:

$$f(X_{1:L} = w_{\widehat{\mathbf{v}}_T[1]} \dots w_{\widehat{\mathbf{v}}_T[L-1]} w_{\widehat{\mathbf{v}}_T[L]} \mid X_{\leq 0})$$
$$\overset{(i)}{=} g(X_L = w_{\widehat{\mathbf{v}}_T[L]} \mid X_{1:L-1} = w_{\widehat{\mathbf{v}}_T[1]} \dots w_{\widehat{\mathbf{v}}_T[L-1]}, X_{\leq 0}) \cdot f(X_{1:L-1} = w_{\widehat{\mathbf{v}}_T[1]} \dots w_{\widehat{\mathbf{v}}_T[L-1]} \mid X_{\leq 0})$$
$$\overset{(ii)}{=} g(X_L = w_{\widehat{\mathbf{v}}_T[L]} \mid X_{1:L-1} = w_{\mathbf{v}_T^*[1]} \dots w_{\mathbf{v}_T^*[L-1]}, X_{\leq 0}) \cdot f(X_{1:L-1} = w_{\mathbf{v}_T^*[1]} \dots w_{\mathbf{v}_T^*[L-1]} \mid X_{\leq 0})$$
$$\overset{(iii)}{>} g(X_L = w_{\mathbf{v}_T^*[L]} \mid X_{1:L-1} = w_{\mathbf{v}_T^*[1]} \dots w_{\mathbf{v}_T^*[L-1]}, X_{\leq 0}) \cdot f(X_{1:L-1} = w_{\mathbf{v}_T^*[1]} \dots w_{\mathbf{v}_T^*[L-1]} \mid X_{\leq 0})$$
$$\overset{(iv)}{=} f(X_{1:L} = w_{\mathbf{v}_T^*[1]} \dots w_{\mathbf{v}_T^*[L-1]} w_{\mathbf{v}_T^*[L]} \mid X_{\leq 0}), \tag{11}$$

where $(i)$ and $(iv)$ follow Assumption 3.3, $(ii)$ corresponds to the setting in this opposite scenario, and $(iii)$ follows Definition 3.1 and that $w_{\widehat{\mathbf{v}}_T[L]} \neq w_{\mathbf{v}_T^*[L]}$. This preference relation in Equation (11) contradicts with that in Equation (9), and therefore, Equation (10) has to hold true.

Therefore, when the advantage of the globally optimal response does not manifest itself until step $L$, the stepwise optimal response deviates from the globally optimal counterpart at some step $K < L$, and that under Assumption 3.4, the following strict preference relations hold true:

$$f(X_{1:K} = w_{\widehat{\mathbf{v}}_T[1]} w_{\widehat{\mathbf{v}}_T[2]} \dots w_{\widehat{\mathbf{v}}_T[K]} \mid X_{\leq 0}) > f(X_{1:K} = w_{\mathbf{v}_T^*[1]} w_{\mathbf{v}_T^*[2]} \dots w_{\mathbf{v}_T^*[K]} \mid X_{\leq 0}),$$
$$\dots \tag{12}$$
$$f(X_{1:L-1} = w_{\widehat{\mathbf{v}}_T[1]} w_{\widehat{\mathbf{v}}_T[2]} \dots w_{\widehat{\mathbf{v}}_T[L-1]} \mid X_{\leq 0}) > f(X_{1:L-1} = w_{\mathbf{v}_T^*[1]} w_{\mathbf{v}_T^*[2]} \dots w_{\mathbf{v}_T^*[L-1]} \mid X_{\leq 0}).$$

This, together with Equation (9) and Assumption 3.3, indicates that:

$$g(X_L = w_{\hat{\mathbf{v}}_T[L]} \mid X_{1:L-1} = w_{\hat{\mathbf{v}}_T[1]} \dots w_{\hat{\mathbf{v}}_T[L-1]}, X_{\leq 0})$$

$$\overset{(i)}{=} \frac{f(X_{1:L} = w_{\hat{\mathbf{v}}_T[1]} \dots w_{\hat{\mathbf{v}}_T[L-1]} w_{\hat{\mathbf{v}}_T[L]} \mid X_{\leq 0})}{f(X_{1:L-1} = w_{\hat{\mathbf{v}}_T[1]} \dots w_{\hat{\mathbf{v}}_T[L-1]} \mid X_{\leq 0})}$$

$$\overset{(ii)}{<} \frac{f(X_{1:L} = w_{\mathbf{v}_T^*[1]} \dots w_{\mathbf{v}_T^*[L-1]} w_{\mathbf{v}_T^*[L]} \mid X_{\leq 0})}{f(X_{1:L-1} = w_{\hat{\mathbf{v}}_T[1]} \dots w_{\hat{\mathbf{v}}_T[L-1]} \mid X_{\leq 0})} = \epsilon_L \tag{13}$$

$$\overset{(iii)}{<} \frac{f(X_{1:L} = w_{\mathbf{v}_T^*[1]} \dots w_{\mathbf{v}_T^*[L-1]} w_{\mathbf{v}_T^*[L]} \mid X_{\leq 0})}{f(X_{1:L-1} = w_{\mathbf{v}_T^*[1]} w_{\mathbf{v}_T^*[2]} \dots w_{\mathbf{v}_T^*[L-1]} \mid X_{\leq 0})}$$

$$\overset{(iv)}{=} g(X_L = w_{\mathbf{v}_T^*[L]} \mid X_{1:L-1} = w_{\mathbf{v}_T^*[1]} \dots w_{\mathbf{v}_T^*[L-1]}, X_{\leq 0}) \leq 1,$$

where $(i)$ and $(iv)$ follow Assumption 3.3, $(ii)$ follows Equation (9), and $(iii)$ follows Equation (12).

Therefore, the conditional probability of generating any $w_{v_L}$ is strictly smaller than a positive number $\epsilon_L$, which is further strictly smaller than a positive number upper-bounded by 1. □

## B  ADDITIONAL RESULTS AND ANALYSES

In this section, we showcase several quantitative experiments for hyper parameter analysis. We also provide several qualitave results to demonstrate the generation process and overall performance of our proposed method.

### B.1  PERFORMANCE ANALYSIS ON MMLU SOCIAL SCIENCE

We further evaluate our method on the social science subset of MMLU using Mistral-Nemo as the base model. The full result is placed in Tab A2. With window size $d = 4$ and threshold $\sigma = 0.5$, our approach demonstrates consistent improvements over baseline decoding methods across most categories. Our method achieves the highest macro average performance compared to greedy decoding and beam search. This consistent performance advantage aligns with our main findings across different model architectures, demonstrating that our method's benefits extend beyond the primary LLaMA-3.5-8B experiments discussed in the main text. The results suggest that our approach's effectiveness in improving model predictions generalizes well across different model families and evaluation scenarios.

Table A2: Performance Comparison of Different Methods by Mistral-Nemo on MMLU Social Science

| Category | Greedy | Beam Search | Ours |
|---|---|---|---|
| Econometrics | 53.51 | 56.14 | **59.65** |
| High School Geography | 78.79 | 77.78 | **79.80** |
| High School Government and Politics | 87.56 | 87.05 | **88.08** |
| High School Macroeconomics | **74.61** | 74.09 | 73.83 |
| High School Microeconomics | 71.24 | **73.39** | **73.39** |
| High School Psychology | **67.16** | 66.01 | 66.50 |
| Human Sexuality | 66.41 | 64.89 | **70.99** |
| Professional Psychology | **67.16** | 66.01 | 66.50 |
| Public Relations | 59.26 | **63.89** | 56.48 |
| Sociology | 76.62 | **82.09** | 74.63 |
| Security Studies | 57.38 | 54.10 | **59.02** |
| US Foreign Policy | **77.00** | 76.00 | **77.00** |
| Macro Average (%) | 71.93 | 72.10 | **72.20** |

### B.2  ANALYSIS ON WINDOW SIZE

We conduct comprehensive experiments on MT-Bench to analyze the impact of window size using both LLaMA-3.1-8B (See Tab A3) and Mistral-Nemo models (See Tab A4), with threshold $\sigma = 0.5$

fixed. For LLaMA-3.1-8B, our method consistently outperforms both greedy decoding and beam search across different window sizes, with $d = 5$ achieving the best overall performance compared to greedy decoding and beam search. For Mistral-Nemo, the optimal performance is achieved at $d = 3$ with an overall score of 7.93, surpassing both greedy decoding and beam search. While $d = 4$ may not always yield the absolute best performance, it demonstrates robust performance across both models and serves as a reliable default setting. These GPT-4o evaluator scores on MT-Bench provide additional evidence supporting our main findings: our method consistently outperforms traditional decoding approaches, confirming its effectiveness beyond the human evaluations presented in the main text.

Table A3: Performance Comparison of Different Window Sizes with Llama3.1-8B on MT-Bench

| Method | Rating 1 | Rating 2 | Overall Mean |
|---|---|---|---|
| $d = 2$ | 8.29 | 7.09 | 7.69 |
| $d = 3$ | 8.35 | 7.51 | 7.93 |
| $d = 4$ | **8.36** | 7.42 | 7.89 |
| $d = 5$ | 8.31 | **7.62** | **7.97** |
| Greedy Decoding | 8.28 | 7.49 | 7.88 |
| Beam Search | 8.07 | 7.19 | 7.63 |

Table A4: Performance Comparison of Different Window Sizes with Mistral-Nemo on MT-Bench

| Method | Rating 1 | Rating 2 | Overall Mean |
|---|---|---|---|
| $d = 2$ | 8.38 | 7.28 | 7.82 |
| $d = 3$ | **8.44** | 7.42 | **7.93** |
| $d = 4$ | 8.28 | 7.41 | 7.84 |
| Greedy Decoding | 8.38 | 7.29 | 7.83 |
| Beam Search | 8.32 | **7.49** | 7.91 |

We further evaluate different window sizes on MMLU social science tasks using Qwen-2.5 7B. The results in Tab A5 show that our method maintains strong performance across various window sizes ($d = 2$ to $d = 6$), with overall macro averages consistently around 79.7%. While $d = 6$ achieves the highest macro average, $d = 4$ demonstrates comparable performance and maintains better computational efficiency. These results further support our choice of $d = 4$ as a robust default setting, offering a good balance between performance and efficiency across different models and tasks.

Table A5: Ablation Study on Window Size $d$ with Qwen-2.5 7B on MMLU Social Science

| Task | d=2 | d=3 | d=4 | d=5 | d=6 |
|---|---|---|---|---|---|
| Econometrics | 62.28 | 62.28 | **64.91** | 64.04 | **64.91** |
| High School Geography | 85.86 | 84.34 | 86.36 | **87.37** | 85.86 |
| High School Government and Politics | **93.26** | **93.26** | 92.23 | 91.19 | 92.23 |
| High School Macroeconomics | 75.90 | **76.15** | 75.13 | **76.15** | 75.64 |
| High School Microeconomics | 83.61 | **84.03** | 83.61 | 82.77 | 83.19 |
| High School Psychology | 87.89 | 88.07 | 88.07 | 88.07 | **88.26** |
| Human Sexuality | 77.86 | 75.57 | 78.63 | **79.39** | 77.86 |
| Professional Psychology | **73.86** | 73.37 | 73.20 | 72.88 | 73.37 |
| Public Relations | 68.18 | **70.00** | **70.00** | 65.45 | 68.18 |
| Sociology | 71.02 | 73.06 | **73.47** | 72.24 | 73.06 |
| Security Studies | 83.08 | 83.58 | 83.08 | 84.08 | **84.58** |
| US Foreign Policy | 86.00 | 86.00 | 86.00 | 86.00 | **88.00** |
| Macro Average | 79.66 | 79.75 | 79.82 | 79.62 | **79.88** |

## B.3 ANALYSIS ON THRESHOLD

We investigate the impact of threshold $\sigma$ on MMLU social science tasks using Qwen2.5-7B with fixed window size $d = 4$. The results in Tab A6 demonstrate that our method performs robustly across $\sigma$ values ranging from 0.25 to 0.75, with $\sigma = 0.25$ achieving the highest macro average of 80.31%. While our default setting of $\sigma = 0.5$ is not the absolute best in this specific experiment, it maintains strong performance and shows consistent improvements across most tasks, suggesting it serves as a reliable default configuration for general use.

Table A6: Ablation Study of $\sigma$ Values on Qwen-2.5 7B on MMLU Social Science Tasks

| Task | $\sigma$=0.1 | $\sigma$=0.25 | $\sigma$=0.5 | $\sigma$=0.75 | $\sigma$=1.0 |
|---|---|---|---|---|---|
| Econometrics | 62.28 | **64.91** | **64.91** | **64.91** | **64.91** |
| High School Geography | **92.23** | 91.71 | **92.23** | **92.23** | 91.19 |
| High School Government and Politics | **92.23** | 91.71 | **92.23** | **92.23** | 91.19 |
| High School Macroeconomics | 75.13 | **76.67** | 75.13 | 75.90 | 75.90 |
| High School Microeconomics | **84.45** | **84.45** | 83.61 | 83.19 | 83.61 |
| High School Psychology | 87.52 | **88.44** | 88.07 | 88.26 | 88.07 |
| Human Sexuality | 74.05 | 77.86 | **78.63** | 77.10 | 77.10 |
| Professional Psychology | 73.20 | **74.35** | 73.20 | 73.20 | 73.04 |
| Public Relations | 69.09 | **70.00** | **70.00** | 67.27 | 67.27 |
| Sociology | 84.58 | **85.07** | 83.08 | 84.58 | 84.58 |
| Security Studies | 72.24 | 71.43 | **73.47** | 72.24 | 72.24 |
| US Foreign Policy | **86.00** | 85.00 | 85.00 | 85.00 | **87.00** |
| Macro Average | 79.40 | **80.31** | 79.82 | 79.88 | 79.69 |

## B.4 ANALYSIS ON MODIFICATION RATE

Table A6: Average Modification Rate across Different Window Sizes $d$

| Window Size | 2 | 3 | 4 |
|---|---|---|---|
| Modification Rate (%) | 9.60 | 6.02 | **3.70** |

To further understand the computational efficiency of our method, we analyze the modification rate under different window size settings. We selected six college-level subject categories from the MMLU test set (including biology, chemistry, computer science, mathematics, medicine, and physics) for analysis, using the Llama3.5-8B model with a threshold of $\sigma = 0.5$. We chose window size $d$ as the key hyperparameter for analysis because it directly influences the modification rate - which is defined as the ratio of the number of triggered refinements multiplied by the window size to the total length of the output text.

As shown in Table A6 and Figure A7, as the window size increases from 2 to 4, the average modification rate shows a clear downward trend, decreasing from 9.60% to 3.70%. This declining trend indicates that larger window sizes lead to a faster decay in the number of modifications. Notably, across all settings, the modification rate remains below 15%, suggesting that our method maintains the same computational complexity as greedy decoding for the majority of the time. This finding highlights the computational efficiency of our approach: it only invokes beam search to find optimal approximations for subsequences when necessary, while maintaining efficiency comparable to greedy decoding otherwise.

## B.5 DEMONSTRATIVE EXAMPLES ON MT-BENCH

In this subsection, we present three examples selected from MT-Bench to illustrate the generation process by reflection window. The first example is the 28-th test sample from MT-Bench. As is shown in Figure A8, the reflection window is being activated when the auto-regressive decoding generates *"part of the car"*. The reflection window goes back to the beginning of the window and rewrites

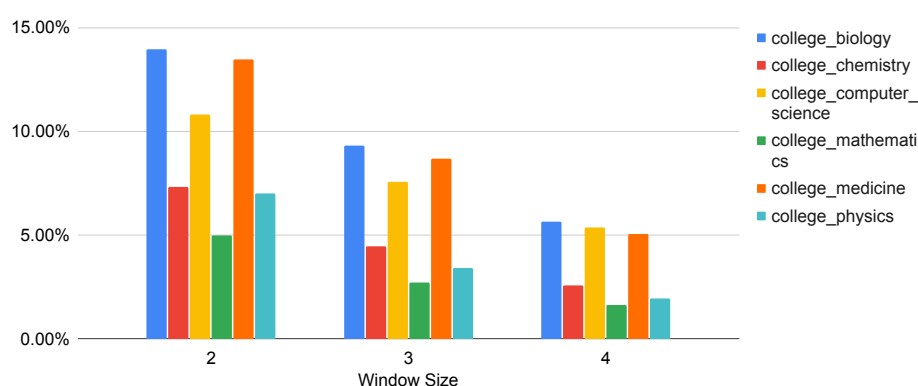

Figure A7: Modification Rate

Which word does not belong with the others?
tyre, steering wheel, car, engine

The word that does not belong with the others is
\"tyre\".
The other three words - \"steering wheel\", \"car\",
and \"engine\" - are all related to a vehicle,
specifically a car.
A \"tyre\", on the other hand, is a component of a
vehicle, but it's a *part of the car* **specific part,
whereas** the other three words are more general
terms related to the vehicle as a
whole.\n\nHowever, if I had to choose one word
that stands out as not belonging, I would say
\"steering wheel\". The other three words - \"car\",
\"engine\", and \"tyre\" - are all nouns that refer to
parts or aspects of a vehicle, whereas \"steering
wheel\" is a specific component of a
vehicle.\n\nBut if I had to choose one word that is
the least related to the others, I would say \"car\".
The other three words - \"steering wheel\",
\"engine\", and \"tyre\" - are all components or
parts of a vehicle, whereas \"car\" is the vehicle
itself.
So, it really depends on how you look at it!

Reflection Window

The word that does not belong with the others is
\"tyre\".
The other three words - \"steering wheel\", \"car\",
and \"engine\" - are all related to a vehicle,
specifically a car.
A \"tyre\", on the other hand, is a component of a
vehicle, but it's a part of the car, rather than the car
itself.

Greedy Decoding

The word that does not belong with the others is
\"tyre\".
The other three words - \"steering wheel\", \"car\",
and \"engine\" - are all related to a vehicle,
specifically a car.
A \"tyre\", on the other hand, is a component of a
vehicle, but it's not a vehicle itself.

Beam Search

Figure A8: Test Sample #28

the part as *"specific part, whereas"* which leads to a more flexible and natural answer. Similarly, Figure A9 and Figure A10 is selected from the 29-th and 7th test sample of MT-Bench, respectively.

One morning after sunrise, Suresh was standing facing a pole. The shadow of the pole fell exactly to his right. Can you tell me the direction towards which the shadow was pointing - east, south, west, or north? …
To which direction was Suresh facing? How do you solve this?

**Reflection Window**

To determine the direction Suresh was facing, we need to consider the position of the sun and the direction of the shadow.
Since the ~~shadow of the pole~~ sun is in the east and its rays cast shadows to the west, and the shadow of the pole fell to Suresh's right, we can infer that Suresh was facing south. This is because, when the sun is in the east, the shadows fall to the west, and if the shadow falls to the right, it means Suresh was facing **south**.

**Greedy Decoding**

To determine the direction in which Suresh was facing, we need to consider the position of the sun and the direction of the shadow.
Since the shadow of the pole fell to Suresh's right, we can infer that the sun is in the east. This is because the sun's rays are coming from the east, and the shadows are cast in the opposite direction.\n\nNow, since the shadow fell to Suresh's right, we can conclude that Suresh was facing the west. This is because the shadow fell to his right, which means he was facing the opposite direction, i.e., the west.\n\nTherefore, the direction in which Suresh was facing is **west**.

**Beam Search**

Since Suresh was standing facing a pole and the shadow of the pole fell exactly to his right, we can deduce the direction of the shadow based on the position of the sun at sunrise.\n\n1. At sunrise, the sun rises in the east.\n2. Since Suresh was facing the pole and the shadow of the pole fell to his right, it means the shadow was pointing in a direction perpendicular to his line of sight.\n3. Given that the sun rises in the east, the shadow of the pole would fall to the right of Suresh's line of sight, which means the shadow would be pointing west.\n\nTherefore, the direction towards which the shadow of the pole was pointing is **west**.

Figure A9: Test Sample #29

Could you write a captivating short story beginning with the sentence: The old abandoned house at the end of the street held a secret that no one had ever discovered.

**Reflection Window**

Here's a captivating short story for you:
The old abandoned house at the end of the street held a secret that no one had ever discovered……
……(omitted)
Suddenly, Emily heard a faint whispering in her ear. It was a soft, raspy voice that seemed to be ~~speaking an ancient~~ coming from all around her. \"Welcome, Emily,\" it said. \"I've been waiting for you.\"
……(omitted)
From that day on, Emily returned to the old house again and again, learning the secrets of the nexus point and the magic that lay within. And as she did, she discovered that the house was not just a place \u2013 it was a doorway to a whole new world of wonder and possibility.

**Greedy Decoding**

Here's a captivating short story for you:
The old abandoned house at the end of the street held a secret that no one had ever discovered……
……
Suddenly, Emily heard a faint whispering in her ear. It was a soft, raspy voice that seemed to be speaking an ancient language she couldn't quite understand. But as she listened more closely, she began to pick out words \u2013 words that seemed to be calling to her, drawing her deeper into the heart of the house.

……(omitted)

But as she looked up, she saw that she was no longer alone. A figure stood in the doorway, watching her with eyes that seemed to hold a deep sadness and a hint of longing. And as Emily looked into those eyes, she knew that she had stumbled upon a secret that was not just about the house, but about herself \u2013 a secret that would change her life forever.

**Beam Search**

Here's a captivating short story for you:
The old abandoned house at the end of the street held a secret that no one had ever discovered. For as long as anyone could remember, the house had stood vacant, its grandeur and beauty slowly being consumed by the passing of time.
…… (omitted)
It was then that Emily realized that she was not just a curious adventurer, but a key player in a much larger game. The old abandoned house at the end of the street was not just a secret, but a doorway to a world of wonder and discovery, and Emily was about to embark on a journey that would change her life forever.

Figure A10: Test Sample #7

# C   PERFORMANCE ACROSS ALL SUBSETS OF MMLU

Table A6: Model Accuracy by Subject Category on MMLU Using Different Methods

| Category | Greedy (%) | Beam Search (%) | Ours (%) |
|---|---|---|---|
| abstract_algebra | 47.00 | 46.00 | 47.00 |
| anatomy | 67.40 | 74.07 | 72.59 |
| astronomy | 80.92 | 73.02 | 80.26 |
| business_ethics | 66.00 | 68.00 | 64.00 |
| clinical_knowledge | 76.22 | 76.60 | 78.11 |
| college_biology | 83.33 | 80.55 | 79.86 |
| college_chemistry | 56.00 | 53.00 | 57.00 |
| college_computer_science | 55.00 | 63.00 | 58.00 |
| college_mathematics | 35.00 | 47.00 | 36.00 |
| college_medicine | 69.36 | 65.89 | 68.78 |
| college_physics | 64.70 | 69.61 | 61.76 |
| computer_security | 73.00 | 78.00 | 77.00 |
| conceptual_physics | 68.93 | 69.78 | 69.78 |
| econometrics | 56.14 | 58.77 | 57.89 |
| electrical_engineering | 58.62 | 64.82 | 61.37 |
| elementary_mathematics | 88.09 | 88.62 | 89.41 |
| formal_logic | 50.00 | 50.79 | 51.58 |
| global_facts | 43.00 | 45.00 | 50.00 |
| high_school_biology | 79.03 | 80.96 | 80.32 |
| high_school_chemistry | 62.07 | 67.98 | 63.05 |
| high_school_computer_science | 81.00 | 79.00 | 79.00 |
| high_school_european_history | 76.36 | 75.15 | 75.15 |
| high_school_geography | 78.79 | 80.81 | 78.79 |
| high_school_government_and_politics | 88.60 | 86.53 | 88.08 |
| high_school_macroeconomics | 69.74 | 74.62 | 68.97 |
| high_school_mathematics | 64.44 | 68.89 | 68.52 |
| high_school_microeconomics | 75.63 | 76.05 | 77.33 |
| high_school_physics | 54.97 | 54.97 | 57.62 |
| high_school_psychology | 85.32 | 88.07 | 86.24 |
| high_school_statistics | 62.96 | 60.19 | 62.96 |
| high_school_us_history | 83.33 | 80.88 | 80.39 |
| high_school_world_history | 78.90 | 80.59 | 76.79 |
| human_aging | 63.68 | 66.82 | 62.78 |
| human_sexuality | 52.67 | 60.31 | 53.44 |
| international_law | 80.17 | 80.17 | 77.69 |
| jurisprudence | 72.22 | 74.07 | 76.85 |
| logical_fallacies | 78.53 | 78.53 | 79.14 |
| machine_learning | 56.25 | 61.61 | 53.57 |
| nutrition | 73.86 | 75.82 | 76.47 |
| us_foreign_policy | 84.00 | 80.00 | 81.00 |
| public_relations | 64.55 | 66.36 | 65.45 |
| world_religions | 83.04 | 79.53 | 82.46 |
| professional_accounting | 51.42 | 53.98 | 51.06 |
| security_studies | 62.45 | 66.12 | 63.27 |
| professional_law | 49.86 | 51.54 | 53.50 |
| professional_psychology | 68.63 | 68.95 | 68.46 |
| prehistory | 70.37 | 72.22 | 70.37 |
| professional_medicine | 79.41 | 79.78 | 79.78 |
| sociology | 74.63 | 77.11 | 73.63 |
| philosophy | 73.06 | 68.49 | 71.40 |
| virology | 48.19 | 47.59 | 49.40 |
| marketing | 88.03 | 84.18 | 87.17 |
| medical_genetics | 77.00 | 79.00 | 75.00 |
| miscellaneous | 86.97 | 87.73 | 87.61 |
| moral_disputes | 67.34 | 66.76 | 64.16 |
| moral_scenarios | 45.47 | 48.72 | 47.03 |

