# OpenReview forum: "Reflection Window: Text Generation with Selective Refinement"
_ICLR.cc/2025/Conference — Submitted to ICLR 2025_

### Official Review · Reviewer_MDmU · 2024-10-27

**Soundness:** 3
**Presentation:** 3
**Contribution:** 2
**Rating:** 5
**Confidence:** 4

**Summary:**

This paper targets to address a limitation of auto-regressive text generation in LLMs, where the process generates text token by token without refinement. They propose a novel method, reflection window, which allows the generation process to pause and reflect on previously generated tokens to correct errors when needed based on an entropy-based criterion. They focus on two benchmarks, MMLU and MT-bench and showed effectiveness compared to greedy decoding, obtaining scores comparable to beam search.

**Strengths:**

The proposed method, reflection window, is novel to address auto-regressive limitations since it includes self-correction in generation steps without generating full token sequences.

**Weaknesses:**

The pausing criterion's dependency on entropy threshold and window size means performance may vary with task and domain shifts. Therefore, it is necessary to consider diverse datasets with various generation tasks.

To demonstrate the robustness and effectiveness of the proposed method, more recent baselines for generation methods need to be considered.

Relying solely on automatic evaluation does not guarantee improvements in fluency, coherence, or error correction.

The paper specifically discusses a few decoder-only LLMs. Different types of models need to be evaluated for robustness.

**Questions:**

.

---

> ### Author Response · Authors · 2024-11-26
> **Response to Reviewer MDmU**
>
> We are very grateful for the thoughtful comments, as well as the time and effort devoted! We have provided a **revised manuscript**, where we use blue font to indicate added/revised material. Below please also see our responses to specific comments and questions:
>
> ---
>
> ### **C1:** "The pausing criterion's dependency on entropy threshold and window size means performance may vary with task and domain shifts. Therefore, it is necessary to consider diverse datasets with various generation tasks."
>
> **A1:** Thanks for the constructive suggestion. In light of your comment, in our revised manuscript, we have included extensive experiments on the MMLU dataset (among other analyses), which contains questions from different categories, e.g., STEM, social science, and so on. The material can be found in Appendices B and C.
>
> ---
>
> ### **C2:** "More recent baselines for generation methods need to be considered"
>
> **A2:** Thanks for the thoughtful comment. By "more recent baselines", we assume we are talking about top-p/k sampling, prompting-based or post-editing approaches. Below we respond in twofold:
>
> 1. Our approach is at the logits level, which is at a different level compared to prompting-based post-editing approaches. In other words, these methods themselves are autogressive text generations that emphasize more on the high-level behavior. These approaches and ours do not replace the function of each other, and can be utilized together.
>
> 1. When demonstrating the potential sub-optimality of purely autoregressive in terms of joint probability over generated tokens, we compare with greedy sampling and beam search, instead of top-p or top-k sampling. The reason is that the greedy sampling corresponds closely with local optimal (for the current step) while the beam search corresponds closely to the global optimal (over several steps), and we intentionally restrict the level of randomness during the process for clearer comparison.
>
> Please kindly let us know if we accidentally misunderstood the suggestion.
>
> ---
>
> ### **C3:** "Relying solely on automatic evaluation does not guarantee improvements in fluency, coherence, or error correction"
>
> **A3:** Thanks for sharing the insights. We totally agree, and this is exactly why in addition to entropy and likelihood evaluations, we also utilized the LLM-as-a-judge protocol, where the benchmark is designed to be better aligned with human preferences (Zheng et al., 2023).
>
> ---
>
> ### **C4:** "The paper specifically discusses a few decoder-only LLMs."
>
> **A4:** Thanks for your comment. We conducted our experiments with decoder-only models, including Llama3.5-8B, Qwen2.5-14B, Phi3-medium and Mistral-Nemo. We focused on decoder-only models because they are widely used in autoregressive text generation tasks, which aligns with the scope of our study.
>
> ---
>
> ### References
>
> Zheng, Lianmin, Wei-Lin Chiang, Ying Sheng, Siyuan Zhuang, Zhanghao Wu, Yonghao Zhuang, Zi Lin et al. "Judging llm-as-a-judge with mt-bench and chatbot arena." Advances in Neural Information Processing Systems 36 (2023): 46595-46623.

---

> ### Author Response · Authors · 2024-12-02
> **Looking Forward to Feedback from Reviewer MDmU**
>
> Dear `Reviewer MDmU`,
>
> We are very grateful for the thoughtful comments, as well as the time and effort devoted. As the discussion phase quickly approaching an end, we are eager to know if our **point-by-point responses**, **further discussions**, and **extensive additional experiments** help address the questions and concerns.
>
> Thanks in advance for engaging in a conversation and sharing your feedback.
>
> Yours sincerely,
>
> Authors of `Submission 8707`

---

### Official Review · Reviewer_DVZQ · 2024-11-03

**Soundness:** 3
**Presentation:** 3
**Contribution:** 1
**Rating:** 5
**Confidence:** 4

**Summary:**

The paper proposes a novel generation technique that allows the LLM to pause autoregressive generation at one point and “reflect” over a window of the generated context, before resuming autoregressive generation. The authors formally show that autoregressive generation is suboptimal even with a good base LLM. Empirically, their approach operates by observing the entropy of previously generated tokens up to a certain window size, and if the entropy is above a certain threshold (indicating uncertainty) then generation is paused and beam search is used instead of greedy decoding. They evaluate their approach over MMLU and MT-bench, comparing it to greedy and beam search.

**Strengths:**

* The theoretical treatment of autoregressive sub-optimality is valid and interesting.
* The authors did a good job highlighting a prevalent issue with current LLMs: The inflexibility of autoregressive word-by-word style generation.
* The proposed approach shows improvement over vanilla greedy decoding.

**Weaknesses:**

* The authors refer to this approach as a "reflection/refinement" technique, but the refinement they refer to merely involves using beam search as opposed to greedy. In other words, the approach seems to me like a hybrid greedy/beam approach, rather than a refinement/reflection setup. While the idea of pause-and-reflect is very interesting, I find the execution to be very poor. Why not pause and run some refinement over the window (e.g., ask the LLM to revise the output).
* The whole approach is based on an assumption that the LLM is calibrated--and therefore an "oracle"---LLM, in the sense that highly likely sequences are correct/preferable. This motvates their assumption that beam search should serve as an approximation to globally optimal sequences. We know this is not the case, and LLMs are in many cases poorly calibrated. This can explain some results where beam search performs worse/on par with greedy decoding (such as in Table 3). In other words, the approach relies on a perfecly calibrated LLM, which may not be available
* Experimental results are not very convincing. Table 1 shows beam search to be better, and only 80 responses were used for evaluation on MT-bench.
* The authors do not provide qualitative examples at all. I would be curious to see how this “refinement” process works, and how the rewritten parts eliminate mistakes and/or improves writing.

Therefore, in its current state, I believe the paper is missing a lot, and I would ask the authors to revise their paper accordingly.

**Questions:**

* After the refinement, what if the entropy condition still holds i.e., the newly generate tokens are also uncertain?

---

> ### Author Response · Authors · 2024-11-26
> **Response to Reviewer DVZQ**
>
> Thanks for the thoughtful and detailed comments, as well as the time and effort devoted! We have provided a **revised manuscript**, where we use blue font to indicate added/revised material. Below please also see our responses to specific comments and questions:
>
> ---
>
> ### **C1:** "The approach seems like a hybrid greedy/beam approach, rather than a refinement/reflection setup [...] Why not pause and run some refinement over the window (e.g., ask the LLM to revise the output)"
>
> **A1:** Thanks for the thoughtful comment. Please allow us to respond in twofold:
>
> 1. At the implementation level, you are right that the approach contains greedy sampling and a local beam search. However, finding an appropriate pausing criterion (to switch from the generation mode to reflection and refinement) also matters. We conduct theoretical analysis (Theorem 3.6) to provide a direct hint on pausing criterion w.r.t. when to switch from generation to refinement.
>
> 1. You are totally right, and asking LLM to run refinement over the window is certainly one way. However, such approach emphasizes more on the high-level behavior of LLMs. In comparison, our approach is at the logits level. The two kinds of approaches do not replace the roles of each other and can be employed together. In this paper, we aim to characterize and address the potential sub-optimality of the purely autoregressive generation itself, at the logits level.
>
> ---
>
> ### **C2:** "Beam search serve as an approximation to globally optimal sequences [...] LLMs are in many cases poorly calibrated [...] the approach relies on a perfectly calibrated LLM, which may not be available"
>
> **A2:** Thanks for sharing your insights. We totally agree that using the sequence from beam search may not correspond to the "globally optimal" response, especially when LLMs themselves are not well-calibrated (as you pointed out). This is exactly why we also have empirical evaluations that use LLM as a judge, when comparing the overall quality of text generated by different approaches.
>
> ---
>
> ### **C3:** "Experimental results are not very convincing."
>
> **A3:** Thanks for the comment. In light of it, we have extensively conducted additional experiments and reported the results in Appendices B and C.
>
> ---
>
> ### **C4:** "The authors do not provide qualitative examples at all. [The reviewer] would be curious to see how this 'refinement' process works, and how the rewritten parts eliminate mistakes and/or improves writing"
>
> **A4:** Thanks for asking about qualitative examples. In addition to the concrete example we originally provided in Figure 3, in the revised manuscript, we have also included additional qualitative examples in Appendix B.5.
>
> ---
>
> ### **Q5:** "After the refinement, what if the entropy condition still holds i.e., the newly generate tokens are also uncertain?"
>
> **A5:** Thanks for the thoughtful question. After the refinement is completed in the window, the slow pointer will catch up with the fast pointer, and the generation continues. In other words, the refinement will not get stuck since the triggering of the pausing criterion happens once, and the sliding window moves on after refinement (if needed) is performed.

---

> > ### Comment · Reviewer_DVZQ · 2024-12-01
> >
> > Thank you for the response and the additional experiments and examples. I have decided to raise my score to 5.

---

> > > ### Author Response · Authors · 2024-12-01
> > > **Thank you for your feedback**
> > >
> > > Dear reviewer DVZQ, thank you for your thoughtful feedback and for taking the time to review our responses and additional experiments! We truly appreciate your constructive comments and your updated assessment. If you have any further questions or additional comments about our paper, we would be happy to address them.

---

### Official Review · Reviewer_cJqb · 2024-11-04

**Soundness:** 3
**Presentation:** 4
**Contribution:** 3
**Rating:** 6
**Confidence:** 3

**Summary:**

This paper proposes a novel technique for generation and reflection of LLM models. It utilizes a fast-slow pointer to maintain a slide window, where reflection tokens are generated. Generally speaking, the proposed method is quite interesting, and could be an easy strategy to implement. The proposed strategy would be able to balance the generation and reflection, and experimental results demonstrate its superior performance compared to greedy decoding and beam search.

**Strengths:**

1) Novelty in technique: Utilizing a fast-slow pointer for reflection and generation is quite a technically interesting idea for LLM decoding.

2) Theoretical formalization is good to understand the problem of auto-regressive understanding.

**Weaknesses:**

1) Insufficient baseline methods: I think author should at least compare the proposed method with:
decoding algorithms: top-k/p sampling
Prompting based ‘reflection’ method and automatic post-editing strategy for fair comparison.
Only comparing with beam/search decoding is insufficient.

2) Lack of Clear Demonstration on Distinction. Though an interesting idea, this paper does not highlight the difference between the proposed strategy with other reflection thinking strategies, practically or principally.

3) lack of logical necessity between the theoretical analysis and the proposed specific method. Despite the theoretical analysis provided in this paper, other methods are also applicable within this theoretical framework and can be viewed as specific cases under this analysis framework. Consequently, why propose a fast-slow pointer under such a framework instead of conventional approaches? Why is the proposed method superior under such an analysis framework? This paper does not answer those questions.

**Questions:**

Please refer to the weakness

---

> ### Author Response · Authors · 2024-11-26
> **Response to Reviewer cJqb**
>
> Thank you for the thoughtful questions and comments, and for the time devoted! We have provided a **revised manuscript**, where we use blue font to indicate added/revised material. Below please see our responses to specific points in the review comments:
>
> ---
>
> ### **C1:** "[The reviewer] think author should at least compare the proposed method with: decoding algorithms: top-k/p sampling, prompting based 'reflection' method, and automatic post-editing strategy for fair comparison."
>
> **A1:** Thanks for the constructive comment. Please allow us to respond in twofold:
>
> 1. Our approach is at the logits level, which is at a different level compared to prompting-based reflection and automatic post-editing methods. In other words, these methods themselves are autogressive text generations that emphasize more on the high-level behavior, and can be used together with our approach (at the logits level when sampling), but they do not directly replace the function of each other.
>
> 1. When we design the experiment to demonstrate the potential sub-optimality of purely autoregressive in terms of joint probability over generated tokens, we compare with greedy sampling and beam search (instead of top-k/p sampling). The reason is that the greedy sampling corresponds closely with local optimal (for the current step) while the beam search corresponds closer to the global optimal (over several steps), and we intentionally restrict the level of randomness during the process.
>
> ---
>
> ### **C2:** "Lack of Clear Demonstration on Distinction"
>
> **A2:** Thanks for the thoughtful comment. In light of your comment, in addition to additional experimental evaluations, we have also included further qualitative examples to demonstrate how our approach works in practice. The related material can be found in Appendix B.5.
>
> In terms of other reflection thinking strategies, they emphasize more on the level of high-level behaviors (e.g., self-correction) while remaining autoregressive at the logits level. Our approach operates at the logits level, and can be utilized together with reflection thinking strategies.
>
> ---
>
> ### **C3:** "The logical necessity between the theoretical analysis and the proposed specific method [...] other methods are also applicable within this theoretical framework and can be viewed as specific cases under this analysis framework"
>
> **A3:** Thanks for carefully thinking about the relation between our theoretical analysis and the empirical approach, and for sharing your insights that the implications of our theoretical analysis framework can be multifaceted.
>
> The initial motivation behind our fast-slow pointer mechanism is to facilitate the built-in refinement/correction of generated contents at the logits level. Since this mechanism itself does not offer a pausing criterion, we conduct the theoretical analysis to provide a direct hint on how to design a pausing criterion to switch from generation to refinement -- when the next-token generation shows noticeable drops in confidence.
>
> ---
>
> ### **Q4:** "Why propose a fast-slow pointer under such a framework instead of conventional approaches? Why is the proposed method superior under such an analysis framework?"
>
> **A4:** Thanks for thoughtful questions. By conventional approaches, we assume we are talking about: first adding a prompt to enforce the language model to perform reflection, and then introducing a specific `<reflect>` token during the supervised fine-tuning (SFT) stage and training the model on a dataset that includes wrong-followed-by-correct examples.
>
> We would like to note distinctions between our fast-slow pointer mechanism and the conventional methods mentioned above.
>
> 1. Our approach addresses the problem at the low-level (logits-level) during sampling. Existing methods, including prompt-based approaches and `<reflect>` token-based SFT models, still operate within the framework of autoregressive decoding. In contrast, our method directly addresses issues at the logits level, potentially enabling the model to avoid suboptimal local solutions during the generation process.
>
> 1. Additionally, SFT-based methods require substantial effort of data collection and model training. Meanwhile, prompt-based approaches may not have a precise control over the reflection behavior and reliable metrics to evaluate it. Our approach circumvents these limitations, providing a more direct and efficient solution.
>
> Please kindly let us know if we accidentally misunderstood your questions.

---

> ### Author Response · Authors · 2024-12-02
> **Waiting for Feedback from Reviewer cJqb**
>
> Dear `Reviewer cJqb`,
>
> Thanks again for the thoughtful comments. In addition to the **point-by-point responses**, we have also provided a **revised manuscript with additional discussions and experiments**. As the discussion phase quickly coming to an end, it would be greatly appreciated if you can kindly let us know if our responses help address the questions and concerns. We cherish this opportunity of communication, and we are eagerly looking forward to your feedback.
>
> Many thanks,
>
> Authors

---

> > ### Comment · Reviewer_cJqb · 2024-12-02
> >
> > Sorry for late response. The authors' response addressed some of my concerns. I raised my score

---

> > > ### Author Response · Authors · 2024-12-02
> > >
> > > Thank Reviewer cJqb for getting back to us, and for the encouraging acknowledgement!
> > >
> > > Please feel free to let us know if you would suggest any further changes. We strive to make our contributions clear, transparent, and comprehensive.
> > >
> > > Sincerely,
> > >
> > > Authors

---

### Official Review · Reviewer_D9CM · 2024-11-05

**Soundness:** 2
**Presentation:** 2
**Contribution:** 2
**Rating:** 3
**Confidence:** 4

**Summary:**

The paper proposes a new decoding strategy called reflection window by using beam search at fixed generation window once detected conditional probability drop at specific position. Furthermore, the paper shows the effectiveness and efficiency of the proposed method compared with two baselines: greedy decoding and beam search on MMLU and MT-Bench with selected subsets.

**Strengths:**

1. The theoretical characterization of sub-optimality is reasonable.
2. The proposed method can be a potential solution for the gap between beam search and greedy decoding.

**Weaknesses:**

1. The experimental results are not convincing: 1) the selection of beam size and window size, there is no analysis about how to select them; 2) Only two baselines are used in the experiments while there are many speculative decoding methods; 3) the performance gap between greedy decoding and reflection window is too small in table 1, there is no significant test.
2. There are a lot of cherry-picking implementation details: 1) the gap in STEM of MMLU is relatively bigger then the author chooses three subsets of STEM to conduct later experiments without revealing the reason and the performance on whole set. This is very important since this makes all conclusions afterwards can not stand. 2) table 2 shows there are only 100 test examples for MT-Bench, and there is no greedy decoding in table 2.
3. The necessity of reflection window is not clear and lots of analysis are missing, including the effects of beam size and window size on the whole set of MMLU and MT-Bench, efficiency analysis and so on.

**Questions:**

1. See above
2. Any human alignment results for MT-Bench since you choose LLM to judge ?
3. Why \sigma can be defined as shown in Eq(4) ?
4. One concern is about section 2.1. If we consider the attention distribution of LLMs, it is possible to let (b) become (a), I do not see the logic here and the proposed method also is auto-regressive method.

---

> ### Author Response · Authors · 2024-11-26
> **Response to Reviewer D9CM  [1 out of 2]**
>
> Thanks for the thoughtful comments, as well as the time and effort devoted! We have provided a **revised manuscript**, where we use blue font to indicate added/revised material. Below please also see our responses to specific comments and questions:
>
> ---
>
> ### **C1:** "The selection of beam size and window size"
>
> **A1:** Thanks for carefully considering the experimental results. In Appendix B of the revised manuscript, we have included additional experiments and analyses (e.g., window sizes and thresholds) to demonstrate the influence of beam and window size. The beam size is fixed to 4 for all experiments.
>
> ---
>
> ### **C2:** "Two baselines are used in the experiments while there are many speculative decoding methods"
>
> **A2:** Thanks for the comment. We respond in twofold, (1) the motivation of speculative decoding, and (2) our approach and speculative decoding are orthogonal and can be utilized together:
>
> - Speculative decoding methods are designed for acceleration, by using a smaller LLM for sampling to mimic as if we were sampling from the original LLM. The process is still autoregressive, which, although faster, subjects to the same pitfall of autoregressive sampling itself.
> - Our method aims to address the pitfall of the autoregressive sampling itself, which is orthogonal to the acceleration of sampling. Our approach can be readily applied on the sampling LLM, e.g., when the smaller LLM is utilized in speculative decoding.
>
> ---
>
> ### **C3:** "The performance gap between greedy decoding and reflection window is too small in table 1, there is no significant test"
>
> **A3:** Thanks for the comment. Please allow us to clarify that there is actually no randomness involved in the sampling itself for the setting in Table 1. The reason is that, for a given LLM and the input prompt, the result of greedy decoding is unique (since only the top-probability token will be sampled out autoregressively). Then, the triggering of switching into the refinement mode is also specified by settings (e.g., window size).
>
> In light of your comment, we have conducted additional experiments and provided the results and analyses in Appendix B.
>
> ---
>
> ### **C4:** "The gap in STEM of MMLU is relatively bigger then the author chooses three subsets of STEM to conduct later experiments"
>
> **A4:** Thanks for carefully considering our empirical evaluations. In light of your comment, we have included more comprehensive experimental results in MMLU. We provide the material in Appendices B and C of the revised manuscript.
>
> ---
>
> (continuing)

---

> ### Author Response · Authors · 2024-11-26
> **Response to Reviewer D9CM  [2 out of 2]**
>
> (continued)
>
> ---
>
>
> ### **C5:** "Table 2 shows there are only 100 test examples for MT-Bench, and there is no greedy decoding in table 2"
>
> **A5:** Thanks for the thoughtful comment. In Table 2, we compare the generated response of our approach, and beam search with greedy decoding, respectively. We follow the pairwise comparison pipeline in MT-Bench (Zheng et al., 2023). Since the greedy decoding serves as the shared baseline between our method and beam search, there is no greedy decoding in Table 2. In order to provide a clearer comparison, we choose the single answer grading as evaluation strategy and provide additional results in Appendix B.2.
>
> ---
>
> ### **C6:** "The necessity of reflection window [...] including the effects of beam size and window size on the whole set of MMLU and MT-Bench, efficiency analysis and so on"
>
> **A6:** Thanks for the constructive suggestion. In light of your comment, we have included additional experimental results in Appendices B and C.
>
> ---
>
> ### **Q7:** "Any human alignment results for MT-Bench since you choose LLM to judge?"
>
> **A7:** Thanks for the thoughtful question. One of characteristics of the MT-Bench benchmark is the agreement between LLM judges and human preferences (Zheng et al., 2023). While we did not conduct human alignment evaluations, we strictly followed evaluation protocol specified by the benchmark, which is consistent with the practice in previous literature, see, e.g., Jiang et al. (2024), Abdin et al. (2024).
>
> ---
>
> ### **Q8:** "Why $\sigma$ can be defined as shown in Eq(4)?"
>
> **A8:** If we understood the question correctly, you were referring to the $\sigma$ in Eq(5). Here, $\sigma$ is a hyperparameter indicating how often the refinement is triggered. For instance, when the entropy threshold is set to higher values (more uncertain about the next token), the refinement is triggered less often, compared to a lower value of $\sigma$ (less uncertain about the next token). Please kindly let us know if we accidentally misunderstood the question.
>
> ---
>
> ### **C9:** "If we consider the attention distribution of LLMs, it is possible to let (b) become (a), [the reviewer does] not see the logic here and the proposed method also is auto-regressive method."
>
> **A9:** Thanks for the careful reading. You are right that an autoregressive model could be a universal way of decomposing the joint distribution, _when considering a fixed-length sequence of tokens_. However, this does not contradict with the fact that the autogressive way of generation can be suboptimal, _even if with an oracle LLM_ (as we showed in Theorem 3.6).
>
> The comparison of Figures 1(a) and 1(b) is to provide an intuitive illustration, where the purely autoregressive way does not capture the selection dependence patterns (since autoregressive generation does not edit/refine generated content, which may depend on future tokens). While the language generation is (locally) token-by-token, at a high level, the generation with reflection and refinement is no longer purely autoregressive.
>
> ---
>
> ### References
>
> Abdin, Marah, Jyoti Aneja, Hany Awadalla, Ahmed Awadallah, Ammar Ahmad Awan, Nguyen Bach, Amit Bahree et al. "Phi-3 technical report: A highly capable language model locally on your phone." arXiv preprint arXiv:2404.14219 (2024).
>
> Jiang, Albert Q., Alexandre Sablayrolles, Antoine Roux, Arthur Mensch, Blanche Savary, Chris Bamford, Devendra Singh Chaplot et al. "Mixtral of experts." arXiv preprint arXiv:2401.04088 (2024).
>
> Zheng, Lianmin, Wei-Lin Chiang, Ying Sheng, Siyuan Zhuang, Zhanghao Wu, Yonghao Zhuang, Zi Lin et al. "Judging llm-as-a-judge with mt-bench and chatbot arena." Advances in Neural Information Processing Systems 36 (2023): 46595-46623.

---

> > ### Author Response · Authors · 2024-12-02
> > **Looking Forward to the Feedback from Reviewer D9CM**
> >
> > Dear `Reviewer D9CM`,
> >
> > Thanks again for the time devoted to the reviewing process. As the discussion phase quickly approaching an end, we are eager to understand if our **revised manuscript**, **point-by-point responses**, and **extensive additional experiments as requested** help address the questions and concerns.
> >
> > We are eagerly looking forward to your feedback.
> >
> > Yours sincerely,
> >
> > Authors of `Submission 8707`

---

### Meta-Review · Area_Chair_21sh · 2024-12-19

**Metareview:**

The paper proposes a method for autoregressive text generation that incorporates a sliding "reflection window" and a pausing criterion to address the limitations of purely autoregressive approaches. The paper makes theoretical claims about the suboptimality of autoregressive generation and introduces a selective refinement mechanism, which pauses generation based on an entropy threshold and applies beam search within a fixed window. The authors claim that this approach improves upon greedy decoding and beam search in selected benchmarks, such as MMLU and MT-Bench.

While the theoretical framing of suboptimality and the motivation for a refinement mechanism are clear, significant concerns were raised regarding the experimental design, execution, and the justification of the proposed method. The reviewers appreciated the theoretical insights and the novelty of incorporating a pausing criterion but noted a lack of rigorous comparisons with other decoding strategies such as top-k/p sampling, prompting-based reflection methods, and automatic post-editing techniques. Concerns were raised about the experimental results due to limited baselines, small test sets, cherry-picked subsets, and the absence of qualitative examples. Additionally, the reliance on beam search as an approximation to "global optimality" assumes well-calibrated LLMs, which is a known limitation. Beam search often fails to outperform greedy decoding in practice, undermining the paper's empirical claims. The authors provided additional experiments and clarifications in the rebuttal, but these were not sufficient to alleviate the reviewers' concerns.

There remains a lack of clarity on why the proposed method is superior within the provided theoretical framework and why it specifically addresses the highlighted limitations better than existing techniques. Furthermore, key issues such as the sensitivity to hyperparameters (e.g., entropy threshold and window size) and the absence of diverse evaluation tasks or human evaluation results leave doubts about the robustness and generalizability of the approach. Given the significant weaknesses in experimental validation, clarity, and justification of the method, the paper does not meet the bar for acceptance at ICLR.

**Additional Comments On Reviewer Discussion:**

During the rebuttal period, the reviewers raised several critical points regarding the proposed method, focusing on its theoretical claims, experimental design, and empirical validation. The authors attempted to address these concerns through clarifications, additional experiments, and revised explanations, but significant doubts remained unresolved.

Theoretical Claims and justification: Reviewers questioned whether the proposed method, which uses beam search within a reflection window, constitutes a true “reflection-based” technique, as it primarily switches between greedy decoding and beam search. Reviewer DVZQ pointed out that the method assumes well-calibrated LLMs, which is rarely the case, undermining the validity of beam search as an approximation for global optimality. Reviewer cJqb further argued that the theoretical framework does not clearly justify the specific use of the fast-slow pointer method over other possible refinements. The authors responded that their approach operates at the logits level and provided additional theoretical context, but they did not convincingly demonstrate the necessity or superiority of their method under the proposed framework.

Experimental Weaknesses and Comparisons: Several reviewers emphasized the limited and cherry-picked nature of the experiments. Reviewer D9CM highlighted the lack of diverse baselines, such as top-k/p sampling and prompting-based methods, while Reviewer MDmU noted the dependency on hyperparameters (e.g., entropy threshold and window size), raising concerns about robustness across tasks and domains. Additionally, DVZQ criticized the small evaluation sets and the absence of qualitative examples demonstrating concrete improvements. In their rebuttal, the authors included extended results in the appendix, addressed concerns about hyperparameter sensitivity, and added qualitative examples. However, these additions were not sufficient to fully address the reviewers' concerns.

Empirical Results and Significance: Reviewers found the performance improvements over greedy decoding and beam search to be marginal and inconsistent. DVZQ and D9CM specifically questioned the practical significance of the proposed method given the small performance gaps and its reliance on beam search. While the authors provided additional experiments and clarified that they intentionally avoided top-k/p sampling for clearer comparisons, the justification for this choice did not fully convince the reviewers.

Human Alignment and Evaluation Metrics: Reviewers noted the absence of human evaluation, particularly for MT-Bench, where LLM-as-a-judge protocols were used. MDmU and DVZQ argued that relying solely on automatic evaluations does not guarantee improvements in fluency or coherence. The authors defended this choice, citing alignment between LLM judges and human preferences.

Final Decision: While the rebuttal addressed some concerns, the fundamental issues around the necessity, robustness, and empirical validation of the proposed method persisted. The theoretical framing, though interesting, lacked sufficient justification for the specific implementation. The experimental results remained limited, and key comparisons were missing. Given these significant weaknesses, the reviewers' concerns outweighed the strengths of the paper, leading to the final decision to reject the submission.

---

### Decision · Program_Chairs · 2025-01-22

Reject